Establishment of a ferroptosis-related gene signature for prognosis in lung adenocarcinoma patients

Cai Jingjing 1
Li Chunyan 2
Li Hongsheng 3
Wang Xiaoxiong 1
Zhou Yongchun chungui7625@163.com 3
1 Molecular Diagnostics Center, The Third Affiliated Hospital of Kunming Medical University/Yunnan Cancer Hospital , Kunming , Yunnan , China
2 Department of Head and Neck Surgery Section II, The Third Affiliated Hospital of Kunming Medical University/Yunnan Cancer Hospital , Kunming , Yunnan , China
3 Molecular Diagnostics Center, The Third Affiliated Hospital of Kunming Medical University/Yunnan Cancer Hospital , Kunming , Yunnan , China
Faber Anthony
Electronic publication date: 2021 Aug 6
Publication date: 2021
Volume: 9
Electronic Location ID: e11931
Received 2021 Mar 2; Accepted 2021 Jul 18
Copyright: ©2021 Cai et al.
Copyright year: 2021
Copyright holder: Cai et al.
License: This is an open access article distributed under the terms of the Creative Commons Attribution License, which permits unrestricted use, distribution, reproduction and adaptation in any medium and for any purpose provided that it is properly attributed. For attribution, the original author(s), title, publication source (PeerJ) and either DOI or URL of the article must be cited.
License URL: https://creativecommons.org/licenses/by/4.0/

Keywords: Lung adenocarcinoma, Ferroptosis, Gene signature, Prognosis, Immune status

Funding: National Natural Science Foundation of China 81860513 This work was supported by the National Natural Science Foundation of China (Nos. 81860513). The funders had no role in study design, data collection and analysis, decision to publish, or preparation of the manuscript.

==============================
Objective

Lung cancer is the most common malignancy worldwide and exhibits both high morbidity and mortality. In recent years, scientists have made substantial breakthroughs in the early diagnosis and treatment of lung adenocarcinoma (LUAD), however, patient prognosis still shows vast individual differences. In this study, bioinformatics methods were used to identify and analyze ferroptosis-related genes to establish an effective signature for predicting prognosis in LUAD patients.

Methods

The gene expression profiles of LUAD patients with complete clinical and follow-up information were downloaded from two public databases, univariate Cox regression and multivariate Cox regression analysis were used to obtain ferroptosis-related genes for constructing the prognos tic risk model, AUC and calibration plot were used to evaluate the predictive accuracy of the FRGS and nomogram.

Results

A total of 74 ferroptosis-related differentially expressed genes (DEGs) were identi fied between LUAD and normal tissues from The Cancer Genome Atlas (TCGA) database. A five-gene panel for prediction of LUAD prognosis was established by multivariate regression and was verified using the GSE68465 cohort from the Gene Expression Omnibus (GEO) database. Patients were divided into two different risk groups according to the median risk score of the five genes. Based on Kaplan-Meier (KM) analysi, the OS rate of the high-risk group was markedly worse than that of the low-risk group. We also found that risk score was an independent prognostic indicator. The receiver operating characteristic ROC curve showed that the proposed model had good prediction ability. Gene Ontology (GO) and Kyoto Encyclopedia of Genes and Genomes (KEGG) functional analyses indicated that risk score was prominently enriched in ferroptosis processes. Moreover, at the score of immune-associated gene sets, significant differences were found between the two risk groups.

Conclusions

This study demonstrated that ferroptosis-related gene signatures can be used as a potential predictor for the prognosis of LUAD, thus providing a novel strategy for individualized treatment in LUAD patients.

Introduction

The incidence and mortality of lung cancer rank first in the world (Siegel, Miller & Jemal, 2020). Among all lung cancer subtypes, lung adenocarcinoma (LUAD) accounts for the highest proportion (∼40%) (Testa, Castelli & Pelosi, 2018). At present, LUAD treatments include surgery, radiotherapy, and chemotherapy, as well as targeted therapy and immunotherapy (Eberhardt & Stuschke, 2015). However, regardless of conventional or novel combination therapy, there appears to be no significant improvement in patient prognosis, metastasis risk, or recurrence rate, and five-year survival remains at only 15% (Chen et al., 2019). Therefore, it is crucial to find innovative prognostic models to provide better diagnosis and treatment strategies for LUAD patients.

Scientists have recently discovered a novel type of programmed cell death that differs from apoptosis and cell necrosis, called ferroptosis, which depends on iron ions and reactive oxygen species (ROS) to induce lipid peroxide accumulation (Latunde-Dada, 2017; Stockwell et al., 2017; Conrad et al., 2018). An increasing body of evidence suggests that ferroptosis is involved in the initiation, progression, and suppression of cancer (Fearnhead, Vandenabeele & Vanden Berghe, 2017). Thus, induction of ferroptosis may be an emerging target for the treatment of malignant tumors (Liang et al., 2019; Hassannia, Vandenabeele & Vanden Berghe, 2019), and polyunsaturated fatty acid (PUFA) in phospholipids, redox active iron, and lipid peroxidation (LPO) repair defects, may determine the susceptibility of cancer cells to ferroptosis. Current studies have shown that ferroptosis mainly involves two pathways: GSH / GPX4 pathway and FSP1 / CoQ / NADPH pathway (Dixon et al., 2012). P53 is the most closely related tumor suppressor gene. It cannot only induce apoptosis, but also induce ferroptosis. P53 can inhibit the absorption of cystine by systemxc by inhibiting the transcription of SLC7A11 (Wang et al., 2016), resulting in the inhibition of the GSH / GPx4 pathway, the reduction of cell antioxidant capacity and the occurrence of ferroptosis. Jiang et al. (2015) confirmed that SLC7A11 is a new regulatory target of p53 gene.

Some genes, such as SLC7A11 (Ji et al., 2018), SLC3A2 (Huang et al., 2005), and STYK1 (Lai et al., 2019), are overexpressed in lung cancer cells and mediate the inhibition of ferroptosis. In addition, some lung cancer drugs have been shown to induce ferroptosis. For example, cisplatin is reported to be an inducer of ferroptosis in non-small-cell lung cancer (NSCLC) A549 cells (Guo et al., 2018) and cisplatin and erastin (a type I ferroptosis inducer (FIN)) in combination exhibit synergistic effects on anti-lung cancer cell activity. Sorafenib has also been found to induce ferroptosis in the lung cancer cell line NCI-H460 (Lachaier et al., 2014). In addition, previous studies have found that GPX4 had high activity in cells with epithelial mesenchymal transition related gene expression. When using first-line chemotherapy drugs to treat melanoma, breast cancer and lung cancer cell lines, the remaining drug-resistant cancer cells were found to have stem cell like characteristics, mesenchymal like gene expression characteristics and GPX4 dependent characteristics (Hangauer et al., 2017). However, there are almost no studies on ferroptosis-related genes and LUAD prognosis. In this research, we explored the prognostic role of ferroptosis-related genes in LUAD.

Materials & Methods

Databases

We used the NCI Genomic Data Commons (https://portal.gdc.cancer.gov) for online analysis and visualization of genomic data from The Cancer Genome Atlas (TCGA) to obtain raw data. We downloaded the TCGA-LUAD Htseq_counts.tsv dataset, which contains 528 tumor samples and 57 normal samples, and downloaded related phenotype information (e.g., age, sex, and TNM stage) and corresponding survival information (e.g., survival status and time to latest follow-up). We searched the LUAD gene expression dataset from the Gene Expression Omnibus (GEO; http://www.ncbi.nlm.nih.gov/geo/) database and collected a cohort based on the GPL96 platform, resulting in 441 cancer patients with complete clinical information.

Collection of ferroptosis-related data

FerrDb (http://www.zhounan.org/ferrdb) (Zhou & Bao, 2020) is a manually collected and curated database for the study of markers and regulators of ferroptosis and the association of ferroptosis disease. At present, ferroptosis-related genes (259) have been found and reported in the literature (Table S1).

Identification of differentially expressed genes (DEGs)

The DEGs were obtained using the “limma” software package (Ritchie et al., 2015) and those DEGs with log2FC —≥ 1 and Padj<0.05 were included in subsequent analyses.

Constructing and validating a prognostic ferroptosis-related gene signature

The ferroptosis-related genes associated with prognosis in LUAD patients were identified using univariate Cox regression analysis. Genes with an adjusted P- value of < 0.05 were included for further analysis. The STRING online database (STRING; http://string-db.org/) (v11.0) (Szklarczyk et al., 2019) was used to analyze the interactions among genes with significant differences in univariate analysis, and a protein-protein interaction (PPI) network was constructed. Using Cytoscape (v3.7.1) (Smoot et al., 2011) to further visualize the DEGs, the top ten genes were screened by the degree method using the cytoHubba plug-in. Multivariate Cox regression analysis was used to obtain the gene panel for constructing the prognostic risk model, which was determined as risk score = ∑(Coefi × Expi). According to the median valuescalculated using the R software packages “survminer” and “survival”, the tumor samples were divided into high-risk or low-risk groups. Kaplan–Meier (KM) survival was further plotted using the “survival” package (Ranstam & Cook, 2017). A time-dependent receiver operating characteristic (ROC) curve drawn with the R package “survivalROC” was used to test the accuracy of the model prediction (Szklarczyk et al., 2019)

Nomogram generation

The R package “RMS” was used to draw a compound nomogram based on risk scores and clinicopathological characteristics, and consistency between the predicted and actual results was evaluated using a calibration curve.

Biological pathway of ferroptosis-related prognostic genes

Gene Ontology (GO) and Kyoto Encyclopedia of Genes and Genomes (KEGG) enrichment analyses (Ashburner et al., 2000) of DEGs in the two risk groups were performed to determine their main biological functions. The “clusterProfiler” package with R language was used to generate corresponding files.Single sample gene set enrichment analysis (ssGSEA) and the “gsva” package (Hänzelmann, Castelo & Guinney, 2013) in R were used for scoring each sample in the two risk groups based on 29 immune-related genes (Table S2) sets to explore the correlations among different risk groups and immune status.

Statistical analysis

Univariate Cox regression, multivariate Cox regression, wilcoxon testand log rank tests were used in this study. Statistical analysis and related figures were generated using R software (v3.5.3) (Diboun et al., 2006). In this study, P adj < 0.05 was regarded as statistically significant.

Results

Patient characteristics and DEGs

Our study flow chart is shown in Fig. 1. In total, there were 528 LUAD patients in the TCGA dataset and 441 LUAD patients in the GSE68465 dataset. Their clinical characteristics are shown in Table 1.

Figure 1 Flow diagram of the bioinformatic analysis in this study.

We obtained 5,438 DEGs in the TCGA database according to — log2FC — ≥ 1 and Padj< 0.05. We used Venn online analysis to identify and visualize overlapping DEGs in the two databases and downloaded the Venn diagram (Fig. 2A). The heat map and volcano map (Figs. 2B, 2C) showed 48 up-regulated genes and 26 down-regulated genes.The list of DEGs is shown in Supplement 3.

PPI analysis of DEGs

Using univariate Cox regression analysis, we identified 17 genes related to prognosis in LUAD patients (Fig. 3A), the correlations of which are shown in Fig. 3B. We then imported the above DEGs into the STRING database, and a PPI network was obtained after the lowest confidence was set to 0.015 and genes without interactions were removed (Fig. 3C). Subsequently, we used Cytoscape and cytoHubba to sketch the top 10 genes. SLC7A11, SLC7A11, and GDF15 were determined to be the top hub genes (Fig. 3D).

Table 1 Clinicopathological characteristics of the LUAD patients.

Characteristics	TCGA (n = 501)	GSE68465 (n = 441)	
	Number of cases	%	Number of cases	%	
Age (years)					
≥65	270	53.9	229	51.9	
< 65	231	46.1	212	48.1	
Gender					
Male	232	46.3	222	50.3	
Female	269	53.7	219	49.7	
T					
T1-2	436	87	401	90.9	
T3-4	65	13	40	9.1	
N					
N0	329	65.7	299	59.7	
N1-Nx	172	34.3	142	40.3	
M					
M0	336	67.1	NA	NA	
M1-Mx	165	32.9	NA	NA	
Stage					
Stage 1–2	394	78.6	NA	NA	
Stage 3–4	107	21.4	NA	NA	

Construction and validation of a prognostic model in TCGA and GEO cohorts

We obtained a five-gene panel as a prognostic signature using the multivariate Cox regression model. The forest plot of the five-gene panel is shown in Fig. 4A. The five screened genes were Arachidonate 15-Lipoxygenase (ALOX15), DNA Damage Inducible Transcript 4 (DDIT4), Hepatocyte Nuclear Factor 4 Alpha (HNF4A), Interleukin 33 (IL33), and Growth Differentiation Factor 15 (GDF15). Among them, DDIT4 and HNF4A were highly expressed in tumor tissues, whereas ALOX15, IL33, and GDF15 were lowly expressed (Table 2). Risk score = (−0.061165052 × expression ALOX15) + (0.169594473 × expression DDIT4) + (0.072356312 × expression HNF4A) + (−0.131023567 × expression IL33) + (−0.107882858 × expression GDF15). Therefore, the risk scores of LUAD patients in the two databases were calculated according to the Cox regression model (composed of five genes).

Figure 2 The differentially ferroptosis-related genes of LUAD in the TCGA database and the FerrDb database.

Figure 3 The differentially ferroptosis-related genes of LUAD in the TCGA database and the FerrDb database.

Patients were dichotomized into low-risk score (n = 251) and high-risk score groups (n = 250) based on the median cut-off value (Fig. 5A). Compared with the low-risk group, the high-risk group showed a higher risk of death (Fig. 5C). Based on three-dimensional (3D) principal component analysis (PCA), the prognosis model clearly distinguished the LUAD tumor samples of the two risk groups (Fig. 5E). Furthermore, KM curve analysis showed that the OS of the high-risk group was significantly worse than that of the low-risk group (Fig. 5G, P < 0.001). The clinical correlation heat map of the two risk groups is shown in Fig. 5I. A time-dependent ROC curve was used to test the predictive performance of the OS risk score composed of the five-gene panel. Results showed that the AUCs of one-year, two-years, and three-years were 0.704, 0.668, and 0.693, respectively (Fig. 5K).

Figure 4 Forest plot of multivariate Cox proportional hazards regression analysis of overall survival for 5-gene signature model in the TCGA cohort.

We also observed similar survival curves, survival statuses, risk scores, patient distributions, and clinical prognostic characteristics in the validated GSE68465 dataset (Figs. 5B, 5D, 5F, 5H, 5J). When we performed 1-, 2-, and 3-year ROC curve analyses to assess the predictive capacity of the prognostic five-gene signature, we found the AUCs for 1-, 2-, and 3-year OS predictions were 0.653, 0.644, and 0.617, respectively (Fig. 5L).

Nomogram establishment

Based on the two cohorts, we constructed a nomogram using clinicopathological characteristics (age, sex, TNM stage, grade) and risk scores, respectively, and calculated the total score of each patient to predict the one-year, two-year, and three-year survival rates of the LUAD patients (Fig. 6A). Further calibration curves showed that the third year OS predicted by the nomogram was in good agreement with actual OS (Fig. 6B).

Assessment of independent prognostic value of risk score

To evaluate the relationship between risk score and prognosis of the five-gene model, we used risk score as an index and clinicopathological characteristics of LUAD patients for univariate and multivariate Cox regression analyses. Available variables of the TCGA cohort included age, sex, TMN stage, and risk score. The GSE68465 cohort included age, sex, TN stage, and risk score. In the TCGA cohort, the risk score was determined as an independent prognostic factor for OS (HR = 1.809, 95% CI = 1.292−2.535, P <  0.001; HR = 1.496, 95% CI = 1.056–2.121, P = 0.023) (Fig. 7A). The GSE68465 cohort results were consistent (HR = 1.625, 95% CI = 1.254–2.105, P <  0.001; HR = 1.425, 95% CI = 1.098–1.850, P = 0.008) (Fig. 7B).

Table 2 The prognostic value of five ferroptosis-related genes.

Gene	Coef	HR	HR.95L	HR.95H	P vaule	
ALOX15	−0.061165052	0.940667967	0.88	1.00	0.049434651	
DDIT4	0.169594473	1.184824275	1.06	1.32	0.002251552	
HNF4A	0.072356312	1.075038325	1.02	1.13	0.003200809	
IL33	−0.131023567	0.877197101	0.79	0.97	0.010695027	
GDF15	−0.107882858	0.897732752	0.83	0.98	0.011022652	
Notes.

Coef is the risk coefficient of each gene, if the value of coef >0, it is regarded as the risk factor of prognosis, otherwise it is regarded as the protective factor of prognosis.

HR hazard ratio

Figure 5 Construction and predictive accuracy in different risk models with TCGA and the GSE68465 dataset.

Figure 6 Construction of survival prediction nomogram and calibration plot of the nomogram.

Figure 7 Forest plot of univariate and multivariate Cox regression analysis in TCGA set (a) and GSE68465 validation set.

Functional enrichment analysis in TCGA and GEO cohorts

We used the R language clusterProfiler package to analyze enrichment in biological functions (GO) and KEGG pathways for the 1,285 DEGs in the high- and low-risk groups in the TCGA cohort. In total, 40 important functional annotations (P < 0.05, Fig. 8A) were enriched in molecular functions, among which five were iron related (P <  0.05, Fig. 8A), including a variety of enzymes, gluconosyltransferase and oxidoreductase activity. Unfortunately, the 16 significantly enriched pathogenic KEGG pathways (P <  0.05, Fig. 8C) were not associated with ferroptosis. In the GSE68465 validation cohort, multiple functional iron-related molecules were enriched (P <  0.05, Fig. 8B). In addition, one iron-related pathway, namely ferroptosis, was enriched in the KEGG validation cohort (P <  0.05, Fig. 8D).

Figure 8 Functional enrichment analysis of two risk groups by Gene Ontology (GO) and Kyoto Encyclopedia of Genes and Genomes (KEGG).

We used ssGSEA to further score the samples from different risk groups in the TCGA and GSE68465 cohorts. Differences in different immune cells, functions, and pathways were detected between the two groups (Figs. 9A–9B). The high-risk group in the TCGA cohort showed lower scores in immune-related cells, such as mast cells, neutrophils, dendritic cells (DCs), and T helper cells, with only natural killer (NK) cells showing higher scores (all adjusted P <  0.05, Fig. 9A). Similar results were obtained for the GSE68465 cohort (all adjusted P <  0.05, Fig. 9B). For pathways, the high-risk group in the two cohorts showed lower scores for type II and type I IFN responses (all adjusted P <  0.05, Fig. 9B).

Figure 9 Comparison of the ssGSEA scores between two risk groups.

Discussion

While the incidence rate of lung cancer is no longer the top priority, the mortality rate is still first among cancer deaths, accounting for more than 180,000 cases [20]. At present, LUAD is still the most common histological subtype of lung cancer, with high mortality among Asians, females, and non-smoking patients (Chen et al., 2020a; Chen et al., 2020b). Although LUAD treatment has made great progress in recent years, most patients with LUAD still exhibit poor prognosis and a low five-year survival rate due to tumor recurrence and metastasis.Therefore, a reliable prognostic biomarker is crucial for evaluating and predicting prognosis in LUAD patients. Recently, bioinformatics analyses have become an important screening tool in cancer research (Huang, Du & Wang, 2019).

Most previous studies have focused on the relationship between ferroptosis and tumorigenesis, development, proliferation, and invasion, with only one study by Liang et al. (2020) exploring ferroptosis-related genes and survival rates in patients with hepatocellular carcinoma (HCC). In this study, we used two independent databases and constructed a ferroptosis-related gene panel to predict OS in patients with LUAD. Firstly, we performed univariate Cox regression analysis on 74 ferroptosis-related DEGs in LUAD patients and found that 17 genes were excellent predictors of prognosis, including seven down-regulated and 10 up-regulated genes. Five ferroptosis-related genes (i.e., ALOX15, ddit4, HNF4A, IL33, and GDF15) were further screened by multivariate regression analysis. We then divided patients into high- and low-risk groups according to their median risk score. KM curve analysis showed that the high-risk group was associated with poorer survival compared with the low-risk group. An ROC curve was used to evaluate the prognostic reliability of this signature. The nomogram results indicated that the risk-score model may be an effective method to predict survival status in LUAD patients over different years. We applied GO and KEGG pathway enrichment analysis and found that risk score was significantly correlated with the biological functions and pathways of ferroptosis. SsGSEA further showed that there were significant differences in the immune status of the DEGs between the high-risk group and low-risk group.

ALOX15 encodes a member of the lipoxygenase family of proteins,which can regulate inflammation and immune responses. Recent studies have shown that ALOX15 is not only involved in apoptosis, but also in autophagy and ferroptosis (Dixon et al., 2012). ALOX15 is involved in ferroptosis through multiple pathways (Stoyanovsky et al., 2019), including regulating the activation of Ras selective lethal small molecular 3 (RSL3) (Probst et al., 2017), reducing the activation of glutathione (GSH), and forming a complex with phosphatidylethanolamine binding protein 1(PEBP1) (Wenzel et al., 2017). Mounting evidence indicates that ALOX15 is down-regulated in many human cancers, including colorectal (Shureiqi et al., 2000), prostate (Tang et al., 2002), breast (Jiang, Douglas-Jones & Mansel, 2003) and lung cancers (Gonzalez et al., 2004). Gonzalez et al. (2004) also reported that the expression of 15-LOX-2 is higher in better differentiated NSCLC and is negatively correlated with tumor grade and tumor cell proliferation. However, whether ALOX15 participates in the occurrence of NSCLC by targeting ferroptosis remains unclear. Previous studies have indicated that the use of 12/15-LOX inhibitors or the silencing of ALOX15 expression can prevent cancer cells (including Calu-1 human NSCLC) with RAS expression from cell death in erastin- and RSL3-induced ferroptosis (Shintoku et al., 2017).

DDIT4 is a stress-response protein whose main function is to inhibit mTOR under stressful conditions.DDIT4 is considered an oncogene (Smith & Xu, 2009), and its overexpression is significantly associated with poorer prognosis in tumor patients (Tirado-Hurtado, Fajardo & Pinto, 2018). Jin et al. (2019) reported that constitutive overexpression of DDIT4 can lead to HSP27 and HSP70 induction and AKT activation. This mechanism is related to lung cancer cell survival and IR resistance, indicating that DDIT4 may be a therapeutic target for lung cancer.

GDF15 is a member of the transforming growth factor-beta superfamily, and its association with cancer can depend on cell state and tumor environment. Recent study suggests that GDF15 is lowly expressed in NSCLC, and its down-regulation is associated with poor prognosis in such patients (Lu et al., 2018). GDF15 is also suggested to inhibit the growth and bone metastasis of LUAD A549 cells by targeting the TGF- β/Smad signaling pathway (Duan et al., 2019). However, GDF15 is up-regulated in some malignant tumors, such as gastric cancer in humans. For example, Chen et al. (2020a) and Chen et al. (2020b) recently reported on the role of GDF15 in gastric cancer cell (MGC803) ferroptosis.GDF15 also plays an important role in erastin-induced ferroptosis by affecting the function of system Xc and regulating the expression of SLC7A11 (Chen et al., 2020a; Chen et al., 2020b). However, the role of GDF15 in LUAD ferroptosis has not yet been reported and needs to be further elucidated.

HNF4A is a nuclear transcription factor, which is highly expressed in most cancers and is significantly associated with poor prognosis. Moreover, HNF4A is reported to play a role in ferroptosis (Dai et al., 2020). Research has indicated that HNF4A expression is up-regulated in HCC (Dai et al., 2020), which can increase the synthesis of GSH and inhibit ferroptosis by up regulating the expression of STMN1 (a ferroptosis down-regulated factor) (Xu et al., 2001). In lung cancer, activation of GSH biosynthesis-related genes can also lead to the inhibition of ferroptosis (Zhang et al., 2019). However, whether and how dysregulated expression of HNF4A regulates GSH production in LUAD remain to be explored.

Martin-Sanchez et al. (2017) showed that IL-33 release is related to ferroptosis, and ferroptosis in acute kidney injury may regulate inflammation by activating IL-33.However, the relationship between ferroptosis in cancer and IL-33 is not clear.Previous studies have reported that CD8 + T cells induce ferroptosis of tumor cells in vivo (Wang et al., 2019; Tang et al., 2020). Our study also explored the relationship between risk score and immune activity, but the mechanism between ferroptosis-related genes of LUAD and tumor immunity needs to be further clarified.

The five genes included in our model play important roles in the occurrence and development of LUAD, and most are related to ferroptosis in malignant tumors. Therefore, this model could be a useful prognostic indicator of LUAD. However, our research has some limitations that most bioinformatics analysis studies share. Firstly, all our data are from public databases, so it will be necessary to verify the prognostic value of the model in clinical samples. Secondly, this study failed to explore the underlying molecular mechanism of ferroptosis-related genes in the occurrence and development of LUAD.

Conclusions

We developed a prognosis signature of five ferroptosis-related genes (ALOX15, DDIT4, HNF4A, IL33, GDF15), which showed good reliability. Analysis of two independent databases demonstrated that this model was independently related to OS in LUAD patients, and thus may be a good predictor of LUAD prognosis.

Supplemental Information

Supplemental Information 1 The relationship between the optimal cut-off expression of each gene and survival prognosis in TCGA cohort

Click here for additional data file.

Supplemental Information 2 The relationship between the optimal cut-off expression of each gene and survival prognosis in GSE68465 cohort

Click here for additional data file.

Supplemental Information 3 259 ferroptosis-related genes

Click here for additional data file.

Supplemental Information 4 The annotated gene set file used in ssGSEA

Click here for additional data file.

Supplemental Information 5 The list of DEGs

Click here for additional data file.

Supplemental Information 6 R files

Click here for additional data file.

Abbreviations

LUAD Lung adenocarcinoma

UCSC The University of California, Santa Cruz

TCGA The Cancer Genome Atlas

GEO Gene Expression Omnibus

DEGs Differentially expressed genes

OS Overall survival

ROC Receiver operating characteristic

GO Gene Ontology

KEGG Kyoto Encyclopedia of Genes and Genomes

SsGSEA Single-sample gene set enrichmentanalysis

3D PCA Principal component analysis

HR Hazard ratio

CI Confidence interval

DCs Dendritic Cells

K–M Kaplan–Meier

ALOX15 The five screened genes were Arachidonate 15-Lipoxygenase

DDIT4 DNA Damage Inducible Transcript 4

HNF4A Hepatocyte Nuclear Factor 4Alpha

IL33 Interleukin 33

GDF15 Growth Differentiation Factor

Additional Information and Declarations

Competing Interests

Author Contributions

Data Availability

The authors declare there are no competing interests.

Jingjing Cai and Chunyan Li conceived and designed the experiments, performed the experiments, analyzed the data, prepared figures and/or tables, authored or reviewed drafts of the paper, and approved the final draft.

Hongsheng Li performed the experiments, prepared figures and/or tables, and approved the final draft.

Xiaoxiong Wang performed the experiments, analyzed the data, prepared figures and/or tables, and approved the final draft.

Yongchun Zhou analyzed the data, authored or reviewed drafts of the paper, and approved the final draft.

The following information was supplied regarding data availability:

The data is available at NCBI GEO: GSE68465 and at the TCGA: TCGA-LUAD.

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
