# Peer review of "Establishment of a ferroptosis-related gene signature for prognosis in lung adenocarcinoma patients"

_PeerJ, doi:10.7717/peerj.11931_

## Round 0.1 · original submission · Major Revisions

Please address the issues raised by the reviewers. In particular, how this study is different from the mentioned study that is quite similar, and some of the same and different conclusions the study may have drawn.

Reviewer 1 ·

Basic reporting

The title, abstract, introduction, methods, results and discussion are appropriate for the content of the text. Furthermore, the article is well constructed, the experiments are well conducted, and analysis is well performed. The figures are relevant, high quality, well labelled and described.

Experimental design

The experimental design is original and the research is within the scope of the journal.. Research question is well defined, relevant and meaningful. The methods are highly technical, ethical and logistical. Statistical methods are chosen correctly.

Validity of the findings

All underlying data have been provided in detail. The findings are meaningful. The conclusions are well stated and relevant to the research questions.

Additional comments

This paper investigates the function of expression of ferroptosis-related genes in overall survival(OS) and prognosis of lung adenocarcinoma(LUAD) by analyzing TCGA datasets. The authors identified a ferroptosis-related gene signature by univariate Cox regression analysis. Moreover, the authors validate their findings in a GEO dataset. To explore further about the related pathways, the authors demonstrate that risk score was prominently enriched in ferroptosis processes. In short, this study identified the ferroptosis-related gene signature as a potential predictor for the prognosis of LUAD.

Editorial Criteria
BASIC REPORTING
The title, abstract, introduction, methods, results and discussion are appropriate for the content of the text. Furthermore, the article is well constructed, the experiments are well conducted, and analysis is well performed. The figures are relevant, high quality, well labelled and described.
EXPERIMENTAL DESIGN
The experimental design is original and the research is within the scope of the journal.. Research question is well defined, relevant and meaningful. The methods are highly technical, ethical and logistical. Statistical methods are chosen correctly.
VALIDITY OF THE FINDINGS
All underlying data have been provided in detail. The findings are meaningful. The conclusions are well stated and relevant to the research questions.

Overall, I think this paper is novel and will be of interest to the community of lung cancer genetics, especially LUAD research. The statistical part is valid and makes sense. The authors make it comprehensive by integrating analysis of multiple sources including GEO and TCGA. The main strengths of this paper is that it addresses an interesting and unexplored question, finds a novel discovery based on a carefully selected set of bioinformatic procedures. As such this article represents an excellent and elegant bioinformatics study which will almost certainly influence our thinking about the function of ferroptosis-related genes in LUAD. Some of the weaknesses are the lack of in vitro or in vivo validation experiments. In general, the work is convincing except some major and minor comments below:


Major Comments:

The Methods of the Abstract are too simple, I would recommend expanding it and adding details like what datasets were used, what statistical methods were used to construct the prediction model, what pathway analysis methods were used. Some contents in the Results could be moved to the Methods.

I do see multiple platforms and normalization methods for LUAD samples in GDC, which includes HTSeq - Counts (sample size: 594), HTSeq - FPKM(sample size: 594) and HTSeq - FPKM-UQ(sample size: 594). Please explain why the sample size in the manuscript is 528 rather than 594. Please also explain why “HTSeq - FPKM” was chosen for analysis, rather than the other two platforms?


I’m wondering if there are any ongoing clinical trials focusing on ferroptosis-related genes identified in this study in LUAD or other types of lung cancer? It will be very strong evidence for the significance of the current study if so.

I’m just wondering if the differences in sequencing platforms affects the results, since the discovery dataset(TCGA) is RNA-seq, and the validation dataset(GEO) is microarray (GPL96 Affymetrix GeneChip Human Genome). I’m wondering if a RNA-seq LUAD dataset as a validation dataset will produce more significant results.





Minor Comments:
Line 58: please replace “UCSC Xena browser (https://xena.ucsc.edu/)” with “NCI Genomic Data Commons (https://portal.gdc.cancer.gov)”. Since GDC is the official data host for TCGA data. Please also add this paper as reference for GDC: https://www.nature.com/articles/s41588-021-00791-5 .

It is great that a session of abbreviations was there to list all the abbreviations for the database names. I would also recommend including abbreviations like LUAD, TCGA, GEO, DEG, GO, KEGG etc in that list.

All the gene names should be italic for all the gene names.

Line 37: please remove the “- -” in the beginning.

Annotated reviews are not available for download in order to protect the identity of reviewers who chose to remain anonymous.

Reviewer 2 ·

Basic reporting

The language used is clear and unambiguous, but introduction and discussion need to be revised by citing the most relevant and recent studies focusing on the correlation between ferroptosis and cancer. For example, cancer cells bearing the mesenchymal phenotype, as well as drug tolerant cancer cells, may be vulnerable to ferroptosis induction ( doi: 10.1038/nature23007, doi: 10.1038/nature24297) and oncogenes can interact with the ferroptotic pathway, rewiring the tumor cells by regulating the expression of the main components of ferroptosis (doi: 10.1158/0008-5472.CAN-20-1641).

Figures 2B, 2C, 3, 5 and 7 are extremely small and their size/resolution has to be improved in order for the specific data to become visible.

Experimental design

The research was within Aims and Scope of the journal and rigorous investigation performed to a high technical & ethical standard.

Validity of the findings

No comments.

Additional comments

In this manuscript Cai et al. elucidate the association between ferroptosis-related genes and prognosis in patients with LUAD (Lung Adenocarcinoma). Specifically, 74 genes involved in ferroptosis mechanisms were found to be differentially expressed between tumor and adjacent normal tissues after interrogating the TCGA and the FerrDb database. A five-gene signature was constructed using multivariate Cox proportional analysis and signature-based risk scores were used to classify patients as high and low risk. Kaplan-Meier survival curves confirmed that Overall Survival was significantly worse in high-risk compared with low-risk patients. Additionally, by performing gene set enrichment analysis and interrogating biochemical pathway databases the authors conclude that the differential expressed genes were mostly enriched in the ferroptosis and the immune-related pathways. In the current study the authors attempt to address an important question by identifying a prognostic marker for lung adenocarcinoma. My specific comments are listed below.

Even though the study is well constructed it lacks novelty, since ferroptosis-related genes have recently been investigated as prognosis prediction markers for lung adenocarcinoma by a recent study by Gao X. et al, (A ferroptosis-related gene signature predicts overall survival in patients with lung adenocarcinoma, Future Medicine, 2021). The genes that were finally used in the two studies for establishing the prognosis signature are different, but the concept, databases, and methodology as well as the conclusions drawn are almost identical.
One additional concern is that most of the genes, that eventually are used as prognostic markers in this study (e.g. DDIT4, IL33), do not constitute the core machinery of the ferroptosis pathway and only indirectly are involved in ferroptosis. SLC7A11 or GCLC that were initially identified as DEGs were not included in the five genes that formed the prognosis signature for LUAD. Based on the five-gene signature created in the current study, could somebody hypothesize that inhibition of ferroptosis is one of the factors involved in the development of lung adenocarcinoma?

Reviewer 3 ·

Basic reporting

No comment

Experimental design

No comment

Validity of the findings

No comment

Additional comments

In this study, Cai et al. proposed a bioinformatics study to find the ferroptosis-related gene signature for
prognosis prediction in lung adenocarcinoma patients. A comprehensive analysis has been conducted and the authors have found 5-gene signature for LUAD prognosis. The manuscript is well-written, there are some major points that can be addressed to improve the study:
1. The authors used datasets from TCGA and GEO also, did they concern about batch effect removal among data?
2. More literature review should be added in terms of bioinformatics analyses on LUAD.
3. Source codes should be provided for replicating the methods.
4. ROC curve has been used in previously bioinformatics studies i.e., PMID: 33539511 and PMID: 33260643. Thus, the authors are suggested to refer to more works in this description.
5. There should have space before "(", i.e., genes(259) ==> genes (259).

---

## Round 0.2 · accepted · Accept

The reviewer is satisfied with the revisions.

Reviewer 2 ·

Basic reporting

The authors have satisfactorily answered the points raised by the reviewers. I have no further comments.

Experimental design

No further comments.

Validity of the findings

No further comments.

Additional comments

The revised manuscript has been improved significantly, and most of my concerns have been addressed.